refugee; posttraumatic stress; cross-cultural; forced displacement; stabilization

**Corresponding author:**
Irja Rzepka-Marot;
Email: irja.rzepka-marot@med.uni-heidelberg.de

# Stabilization interventions in the treatment of traumatized refugees: A scoping review

Irja Rzepka-Marot[1,2] , Nadja Gebhardt[1,2], Jonathan Nowak[1,2], Bastian Bruns[1,2], Hans-Christoph Friederich[1,2] and Christoph Nikendei[1,2]

[1]Department of General Internal Medicine and Psychosomatics, University Hospital Heidelberg, Germany and [2]Deutsches Zentrum für Psychische Gesundheit, DZPG (German Centre for Mental Health – Partner Site Heidelberg/Mannheim/Ulm), Germany

## Abstract

Refugees and forced migrants are particularly susceptible to trauma-related disorders, due exposure to traumatic events before, during or after displacement. In trauma therapy, the concept of psychological stabilization refers to the improvement of a patient's capacity to manage symptoms and emotions associated with traumatic experiences. While exposure-based therapies are widely recommended for treating posttraumatic stress disorder (PTSD), stabilizing interventions may offer a valuable alternative, particularly given the unique challenges in refugee care. This scoping review aims to provide a comprehensive overview of stabilizing, non exposure-based interventions for traumatized refugees A systematic search identified 31 relevant studies featuring diverse interventions, settings, and outcomes. Most studies showed a significant reduction in PTSD symptoms compared to waitlist (six studies), treatment as usual (three studies) and pre-post analyses (nine studies), though nine studies found no difference between intervention and comparison group. Notably, two studies found the stabilizing approach less effective than the comparison group, and two reported no symptom reduction in pre-post analysis. Heterogenity among the examined interventions as well as living conditions was high and limited the generizability of the results. Further studies should take these environmental factors into consideration.

## Impact statement

This scoping review investigates the potential of stabilizing interventions as an alternative approach to exposure-based therapies for treating trauma-related disorders in refugee populations, a group particularly vulnerable to trauma. By offering a comprehensive review of nonexposure-based interventions, this study provides valuable insights into the current state of research on stabilizing interventions for refugees under different living conditions. The 31 studies included in the review were comprised of over 15 different interventions implemented through various formats. They showed mixed results, but most reported a significant reduction in PTSD symptoms compared to waitlist, treatment as usual or in pre–post analyses. These findings highlight the importance of future research that considers the different living conditions of refugees.

## Introduction

By the end of 2023, approximately 117 million individuals worldwide – around 1 % of the global population – were forced to leave their places of residence. This displacement occurred within their country of origin, neighboring countries or countries further away (UNHCR, 2024). As refugees are frequently exposed to traumatic events (Abu Suhaiban et al., 2019; Nesterko et al., 2019; Acarturk et al., 2021), they are disproportionately affected by trauma-related disorders compared to the general population, with most studies reporting a prevalence rate of posttraumatic stress disorder (PTSD) exceeding 30% (Kaltenbach et al., 2018; Blackmore et al., 2020). PTSD is characterized by re-experiencing through intrusions, flashbacks or nightmares, hyperarousal and avoidance of trauma-related stimuli (World Health Organisation, 2020), which can significantly diminish the quality of life (Monson et al., 2017; Lefebvre et al., 2021). Moreover, in refugee populations, mental health impairments are closely associated with integration difficulties (Schick et al., 2016). These reasons call for efficient and timely treatment.

Recent research has demonstrated the efficacy of exposure-based therapy interventions for the treatment of PTSD (McLean et al., 2022). As a result, it is now the standard treatment recommendation in several treatment guidelines (International Society for Traumatic Stress Studies Guidelines Committee, 2018; Hamblen et al., 2019; Schäfer et al., 2019). A defining

characteristic of exposure-based interventions is that patients are confronted with traumatic memories or trauma-related stimuli in a therapeutic manner, with the objective of processing the associated emotions (Rothbaum and Schwartz, 2002). However, exposure-based interventions are also associated with a higher dropout rate than other psychotherapeutic interventions, possibly related to the distress resulting from the confrontation with intense negative emotions (Lewis et al., 2020). Outside of controlled clinical study conditions, various factors on the clinician side, such as concerns about worsening symptoms, or on the patient side, such as comorbidities, also contribute to the infrequent implementation of exposure-based therapies. These factors can lead to the adaptation of manualized interventions to better suit individual cases, rather than being strictly implemented as originally designed (Najavits, 2015). As exposure-based interventions might lead to considerable treatment-associated distress through the reprocessing of traumatic experiences and the accompanying emotions (Foa et al., 2002), an adequate level of psychological stability is warranted. Moreover, the stability of the therapeutic relationship – encompassing both continuity of treatment, particularly during exposure to traumatic memories and the reliability of the therapist–patient bond to prevent therapy discontinuation – is essential in order to successfully conduct exposure-based interventions (Gjerstad et al., 2024).

The prerequisites for exposure-based interventions are particularly hard to fulfill when treating PTSD in refugee populations, as there are many legal and structural obstacles which must be overcome to provide refugees with healthcare (Giacco et al., 2014). In general, the development of a stable therapeutic relationship can be impeded by a high degree of mistrust often displayed by individuals who have lived through interpersonal trauma (Hembree et al., 2003; Olatunji et al., 2009). The language barrier is often identified as a significant challenge in clinical healthcare settings, rendering the establishment of a stable therapeutic relationship more challenging. The lack of interpreters (Bell and Zech, 2009) or inadequate reimbursement policies often impede patients from accessing necessary and adequate treatment of any kind (Helmboldt et al., 2019). Access to adequate mental health care often depends on a person's asylum status (Bell and Zech, 2009). Moreover, the uncertainty regarding legal status and housing, commonly faced by refugees, along with cultural challenges, can threaten psychological stability. This may deter mental health professionals from providing exposure-based interventions, especially when treatment continuity cannot be guaranteed (Bell and Zech, 2009). The systemic challenges of mental health care for refugees apply to all mental disorders and therapeutic approaches. They also have an impact on the implementation and execution of studies, making it challenging to collect data of high quality (Panter-Brick et al., 2020; Hinchey et al., 2023b). Nevertheless, attempts have been made to assess the efficacy of the exposure-based treatments for refugees and asylum seekers, with promising results (Kaltenbach et al., 2020), including narrative exposure therapy (NET) therapy in a refugee camp in Uganda (Neuner et al., 2008) and eye movement desensitization and reprocessing (EMDR) in a refugee camp in Turkey (Acarturk et al., 2015, 2016; Yurtsever et al., 2018). When examining individual studies, the following aspects should be considered: it is important to note that the improvement in PTSD symptoms among participants of NET therapy (Neuner et al., 2008) was reported at the 1-year follow-up, during which most participants in this group no longer resided in the camp, raising questions about whether the symptom improvement was attributable to the intervention or improved living conditions (Mundt et al., 2014). During the follow-up of a group EMDR therapy, no differences were observed between

the intervention and control groups, despite an initial reduction in symptoms in the intervention group. The authors primarily attributed this outcome to the persistently stressful living conditions in the refugee camp (Yurtsever et al., 2018). In a meta-analysis by Turrini et al., no significant effectiveness was found for NET and EMDR in refugee populations (Turrini et al., 2019). This finding contrasts with the results of a meta-analysis conducted by Nosè et al., which specifically investigated psychological interventions for refugees in high-income countries and demonstrated the effectiveness of NET in this setting (Nosè et al., 2017). Nonetheless, it should be noted at this point that a meta-analysis by Turrini et al. also identified trauma-focused cognitive behavioral therapy (CBT) as an effective intervention, with sustained effects at follow-up despite the presence of postmigratory stressors (Turrini et al., 2019). While trauma exposure-based interventions have shown effectiveness, albeit with variable levels of evidence, alternative treatment approaches for situations in which the prerequisites for exposure-based interventions are not met could represent a valuable addition to improving the mental health of refugee populations.

Stabilizing interventions, which are commonly well-established in clinical settings (Rosner et al., 2015; Equit et al., 2018), represent an alternative treatment approach. They can reduce trauma-related symptoms but can also serve as a preparation for exposure-based interventions as part of a phase-based approach (Willis et al., 2023). They are designed to assist trauma survivors in managing trauma-related symptoms without using maladaptive regulation strategies. In psycho-traumatology, the term "stable" is commonly used to describe a person who is capable of coping with trauma-related stimuli, emotions and memories, without risk of serious deterioration in their general physical and mental well-being. This encompasses the absence of behaviors such as self-harm, suicidal ideation, substance abuse and dissociative episodes (Reddemann, 2011). In this context, stability mainly refers to a trauma survivor's inner stability (e.g., coping with symptoms) but also includes external aspects and risk assessments, such as social stability (social support system and network), physical and psychological safety (perpetrator contact, living conditions) and the nature of the patient–therapist relationship (Sack and Gromes, 2013; Seidler et al., 2015). While many definitions of stabilization interventions exist, our review uses the definition proposed by Luise Reddemann, as it aligns with the current clinical understanding of stabilization in treating trauma-related disorders. According to Reddemann, stabilization interventions aim to enhance symptom management, emotion regulation and the acquisition of new competencies (Reddemann, 2011; Reddemann and Piedfort-Marin, 2017). This is achieved through regaining a sense of control (Herman, 1992), interpersonal safety (Willis et al., 2023) and strengthening socio-psychological skills (Ter Heide et al., 2016). Stabilization techniques do not make use of traumatic memories. Thus, psychological stability, treatment continuity and the development of a stable therapeutic relationship are of lesser importance than exposure-based therapy.

The necessity of stabilization interventions is a topic of critical debate. Concerns have been raised that stabilization interventions might delay the implementation of evidence-based exposure-based interventions (Neuner et al., 2008; De Jongh et al., 2016). While many treatment manuals for exposure-based interventions, such as EMDR, incorporate stabilization elements (Foa et al., 2008; Shapiro, 2017), usually evaluations assess the therapy manual as a whole (Rosner et al., 2015). Therefore, the specific impact of the stabilization elements cannot be determined independently from the overall treatment effect and systematic evidence for the efficacy of stabilizing interventions is lacking (National Institute for Health

and Care Excellence, 2018; Berliner et al., 2019). Thus, the aim of this scoping review is to provide an overview of the current research on stabilizing, nonexposure-based interventions for refugees with trauma-related disorders through a systematic search of the literature. In light of the expansive scope of this definition, the review will initially present the interventions conducted in the included studies, followed by an analysis of the study designs, including participants and outcomes.

## Methods

### Search strategy

As this was the first review on stabilizing interventions in refugee populations and considering the large variety of implementations of stabilizing interventions, we decided to conduct a scoping review of the literature. Herein, we adhered to the guidelines laid out by the PRISMA-ScR checklist (Tricco et al., 2018). We conducted a systematic search to identify studies that examine stabilizing interventions and evaluate their effect on the symptom burden of PTSD in adult refugees. With regard to the criterion of whether a study examines a stabilizing approach, we have used the above-mentioned definition by Luise Reddemann (2011) as a guideline. The final decision as to whether an intervention was "stabilizing" was the subject of discussion among the reviewers. The screening process was conducted by three independent reviewers (IR, NG, JN). The search was conducted on PubMed, Embase, Web of Science, PsycInfo and CINAHL.

### Inclusion criteria

Studies were included if the sample was comprised of refugees or forced migrants worldwide at all stages of flight, i.e. internally displaced persons, people in refugee camps or the postmigration phase at various stages of the asylum process. We included intervention studies, such as randomized controlled trials, as well as single interventions from all regions of the world. The respective interventions could be performed by professionals or lay providers. Individual and group interventions were included. Further, we included studies that evaluated the effect of stabilizing interventions on reducing PTSD symptom load, with PTSD as either the primary or secondary outcome. Initially, we planned to include only studies focusing on adult refugees ≥18 years. However, some projects targeted communities, families or "youth," with some individuals being under 18 years. Therefore, we adjusted our criteria to include studies in which the majority of participants were adult refugees. Exclusion criteria were other types of publications such as abstracts, conference papers or dissertations, single case studies, systematic reviews and meta-analyses, studies conducted among only underage refugees or qualitative studies. In addition, we excluded studies that did not provide sufficient details about stabilizing interventions, multimodal interventions and interventions that included trauma exposure elements. No date restriction was placed on this search. Only papers in English or German language were included. Additionally, we conducted a forward reference search to identify further publications on the topic. Search terms related to the population were: refugees OR asylum seekers OR forced migration OR displaced people. Search terms related to the outcome were PTSD or posttraumatic stress disorder OR trauma OR traumatized. Search terms related to therapy were therapy OR intervention OR treatment OR psychotherapy OR stabilization. The full search

**Table 1.** Key search terms for EMBASE

| Population | refugee*, "asylum seeker*," "displaced people", "displaced person*"<br>"displaced population", "forced migra*", refugee/exp., "forced migrant"/exp |
| --- | --- |
| | OR/1–8:ti,ab |
| | AND ([english]/lim OR [german]/lim) |
| Intervention | "therap*," intervention*, treat*, stabiliz*, stabilis*, psychother*, psychosoc*, therapy/exp., intervention/exp |
| | OR/1–9:ti,ab |
| | AND ([english]/lim OR [german]/lim) |
| Outcome | PTSD, "Posttraumatic stress disorder," "Post-traumatic stress disorder," traumatized, trauma, "trauma-affected," Trauma/exp., Psychotrauma/exp., Posttraumatic stress disorder/exp |
| | OR/1–9:ti,ab |

*Note*: Date of search: 10.11.2023.

strategy for EMBASE can be found in Table 1. The protocol of this scoping review can be assessed at OSF (https://osf.io/z3dcy).

### Data extraction and management

For each included study, information on author and year of publication, study design, country of study conduction, study population, sample size and gender distribution, inclusion criteria, information on the stabilizing intervention, PTSD outcome measure and PTSD symptom outcome was collected by the reviewers on a data collection form. The full screening process is displayed in Figure 1.

### Risk of bias assessment

For each included study, we also conducted a risk of bias evaluation. For this purpose, we utilized the Cochrane evaluation tools, specifically the revised RoB 2 tool for randomized controlled trials (Sterne et al., 2019) and the ROBINS-I tool for nonrandomized studies (Sterne et al., 2016).

## Results

### Study selection and procedure

We indentified a total of 5,115 studies after automated removal of duplicats (see Figure 1). Following the screening of titles and abstracts, 5,048 results were excluded. The full text was assessed for eligibility for 67 studies. A total of seven studies were excluded because the intervention included exposure-based elements. Another 11 studies were excluded because the specifications in the study design did not meet the definition of stabilization (Kruse et al., 2009; Renner et al., 2011; Jespersen and Vuust, 2012; Ter Heide and Smid, 2015; Stammel et al., 2017; Yurtsever et al., 2018; Shultz et al., 2019; Trilesnik et al., 2019; Park et al., 2020; Gever et al., 2023; Graef-Calliess et al., 2023). Three studies were excluded because the available information about the examined intervention was insufficient (Neuner et al., 2010; Rees et al., 2013, 2014). With regard to the intervention, six studies lacked any measure of PTSD symptomatology (Renner et al., 2008; Sonne et al., 2016, 2021; Acarturk et al., 2022; Aizik-Reebs et al., 2022; Orang et al., 2022). Three studies could not be found (Renner and Peltzer, 2008; Kayal et al., 2013; Sonne et al., 2019), and six were conference

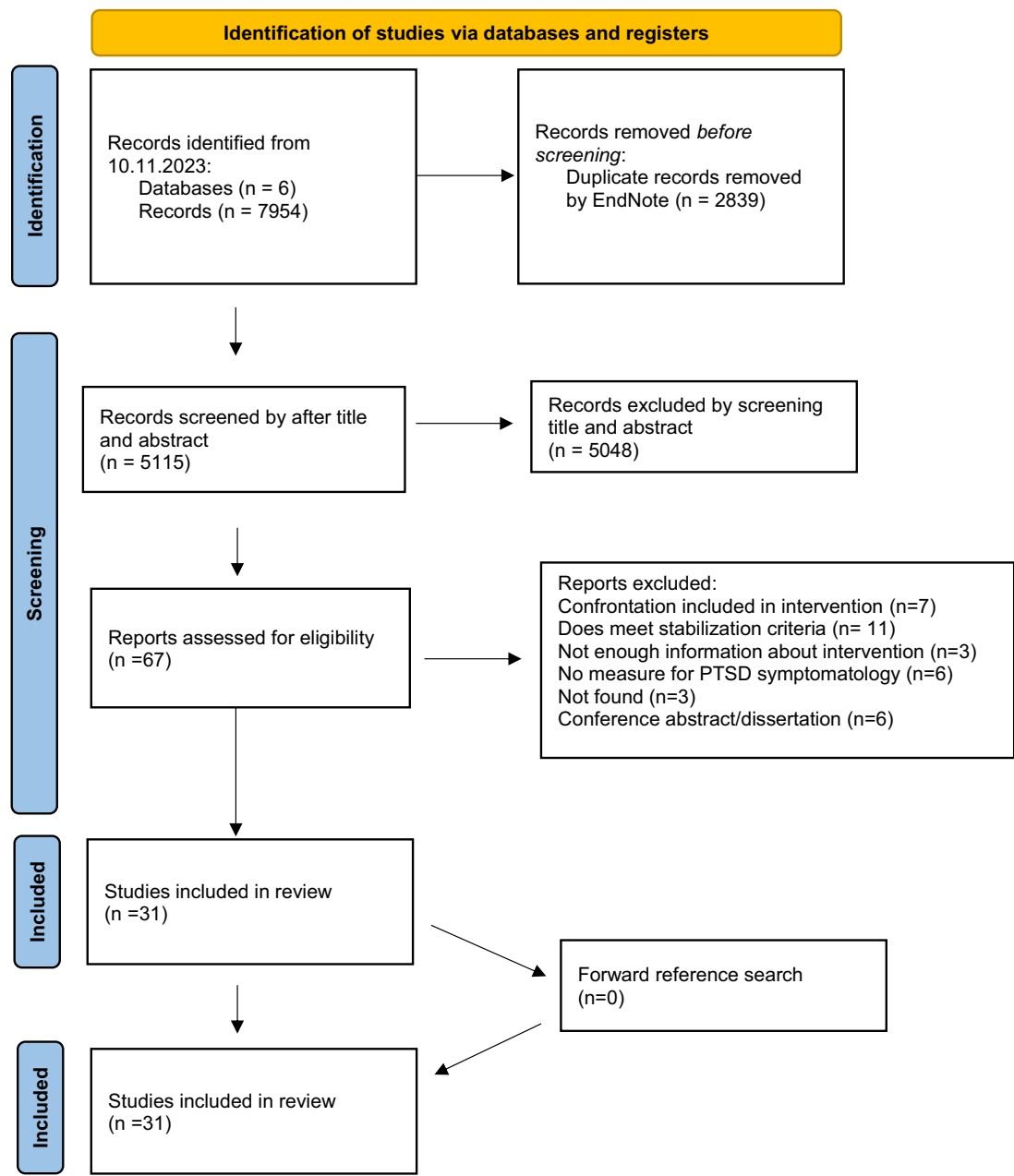

**Figure 1.** PRISMA 2020 flow diagram.

abstracts or nonpeer-reviewed dissertations (Mojica-Castillo, 2003; Stenmark et al., 2008; Ekstrøm et al., 2016; Bryant, 2022; De Graaff et al., 2022; Stöckli et al., 2023). A total of 31 studies were included in our final analysis. An overview of the included studies can be found in Table 2. The 31 studies included a total of 2,759 participants (1,192 male, 1,269 female, 298 not known) receiving a stabilizing intervention. An overview of the included interventions can be found in Table 2.

### Examined stabilizing interventions

The 31 studies analyzed presented a variety of stabilizing treatment approaches. A total of 15 interventions were conducted in a group

format (Yeomans et al., 2010; Stanford et al., 2014; Im et al., 2018; Zehetmair et al., 2018; Alsheikh Ali, 2020; Koch et al., 2020; Lancaster and Gaede, 2020; Tol et al., 2020; Aizik-Reebs et al., 2021; Akhtar et al., 2021; Acarturk et al., 2022; Bryant et al. 2022a,b; Griggs et al., 2022; Hasha et al., 2022), 14 in individual sessions (Neuner et al., 2004; Ter Heide et al., 2011; Hensel-Dittmann et al., 2011; Stenmark et al., 2013; Meffert et al., 2014; Ter Heide et al., 2016; Brakemeier et al., 2017; Altawil et al., 2018; Carlsson et al., 2018; De Graaff et al., 2020; Barhoma et al., 2021; Knefel et al., 2022; Orang et al., 2022; De et al., 2023) and two as smartphone-based interventions (Mazzulla et al., 2021; Röhr et al., 2021).

Seven studies explored the effectiveness of Problem Management Plus (PM+), a transdiagnostic intervention developed by the World

**Table 2.** Overview of included studies

| Author & year of publication | Study design | Country | Participants | Inclusion criteria | Stabilizing intervention | PTSD outcome measure | Outcome for PTSD symptomatology |
|---|---|---|---|---|---|---|---|
| Acarturk et al. (2022) | Pilot RCT with enhanced care as usual | Turkey | Syrian refugees | K10[1] > 15 WHODAS[2] 2.0 > 16 | Group Problem Management Plus (gPM+) | PCL–5[3] | Significant effect of time of both interventions (F(2,88) = 10.01, p = .000, d = .66.) |
| Aizik-Reebs et al. (2021) | RCT with waitlist | Israel | Eritrean refugees | Not specified | Mindfulness-based Trauma Recovery for Refugees (MBTR-R) | HTQ[4] | Significant reduction of PTSD symptoms relative to waitlist group F(1,74) = 12.44, p = .001, $\eta^2$ = .17 |
| Akhtar et al. (2021) | Feasibility RCT with enhanced care as usual | Jordan | Syrian refugees | K10 > 16 WHODAS[2] 2.0 > 17 | Group Problem Management Plus (gPM+) | PCL–5[3] | Descriptive reduction of mean for PTSD symptoms in intervention group (pre: 32.00 (19.21), post: 22.86 (17.00)) |
| Alsheikh Ali et al. (2020) | Quasi experimental RCT with control group | Jordan | Syrian refugee women | PCL–5[3], Ryff's psychological well-being scales (PWB), | Counseling with focus on alienation, loss and grief, hope and hopelessness, and psychological problems | PCL–5[3] | Significant reduction of PTSD symptoms relative to waitlist control group F(1,35) = 6.69, p = .01 |
| Altawil et al. (2018) | Comparison of 3 interventions, non-randomized | Gaza | Palestinian refugees (12–75 yrs) | Participants with symptoms of PTSD" | Community Wellness Focussing (CWP) | PTSD-SRII scale[5] | Reduction of PTSD symptoms in CWF group (t(113) = 23.12, p < 0.01) |
| Barhoma et al. (2021) | RCT with Stress Management vs. Cognitive Restructuring – follow-up | Denmark | Refugees in Denmark | PTSD according to ICD 10 | Stress Management (SM) | HTQ[4] | Nonsignificant interaction effect on PTSD symptoms six (p = .441, d = 0.19) and 18-month posttreatments (p = .724, d = 0.14) |
| Brakemeier et al. (2017) | Feasibility study | Germany | Syrian refugees | Affective/ anxiety/ somatization/ eating disorder, PTSD, substance abuse according to DSM-IV | Interpersonal therapy | PDS[6] | Nonsignificant reduction of mean for PTSD symptoms in pre–post analysis (pre: 30.57 ± 13.29; post: 20 24.1 ± 12.59) |
| Bryant (2022) | RCT with enhanced usual care | Jordan | Syrian refugees | K10[1] ≥ 16 WHODAS 2.0[2] ≥ 17 | Group Problem Management Plus (gPM+) | PCL–5[3] | No significant differences between conditions for changes in PTSD after 3 months (adjusted mean difference − 0.68, 95% CI –4.01 to 2.66; p = 0.69) |
| Bryant (2022) | RCT with enhanced usual care – follow-up | Jordan | Syrian refugees | K10[1] ≥ 16 WHODAS 2.0[2] ≥ 17 | Group Problem Management Plus (gPM+) | PCL–5[3] | No significant differences between conditions for changes in PTSD after 12 months (adjusted mean difference: −0.06 (95% CI −3.48 to 3.36) p = 0.97) |
| Carlsson et al. (2018) | RCT with Stress Management vs. Cognitive Restructuring | Denmark | Refugees in Denmark | PTSD according to ICD 10 | Stress Management (SM) | HTQ[4] | No significant difference of change for PTSD symptom between groups (difference mean pretreatment: −0.03 (0.08), posttreatment = 0.04 (0.11), p = 0.07 (0.09), p = 0.45, d = 0.16 |
| De Graaf et al. (2020) | Pilot RCT with care as usual (CAU) | Nether-lands | Syrian refugees | K10[1] > 15 WHODAS 2.0[2] > 16 | Problem Management Plus (PM+) by peer providers | PCL–5[3] | Significant interaction effect in favor of PM+/CAU between time and condition for symptoms of PTSD (χ2(2) = 9.07; p = 0.010) |
| De Graaf et al. (2023) | RCT with care as usual (CAU) | Nether-lands | Syrian refugees | K10[1] > 15 WHODAS 2.0[2] > 16 | Problem Management Plus (PM+) by peer providers | PCL–5[3] | At 3-month FU, PM+/CAU had greater reductions on PTSD symptoms (p = 0.0005, Cohen's d = 0.39) |

*(Continued)*

**Table 2.** (*Continued*)

| Author & year of publication | Study design | Country | Participants | Inclusion criteria | Stabilizing intervention | PTSD outcome measure | Outcome for PTSD symptomatology |
|---|---|---|---|---|---|---|---|
| Griggs et al. (2022) | Pilot pre–post evaluation | Great Britain | refugees from 22 different countries | Not specified | Manualised stabilization based on CBT (Moving on after trauma, MOAT) | IES-R[7] | Significant reduction of mean for PTSD symptoms in pre–post analysis (p = .001, $\eta p^2$ = .393) |
| Hasha et al. (2022) | RCT with waitlist | Norway | Syrian refugees | IES-R[7] > 24 | Teaching Recovery Techniques (TRT) | IES-R[7] | No significant reduction of mean for PTSD relative to waitlist (−1.3 (−8.7, 6.2)) |
| Hensel-Dittmann et al. (2011) | RCT Narrative Exposure Therapy vs. Stress Inoculation Training | Germany | Not specified | "history of experiencing organized violence, current PTSD diagnosis" | Stress Inoculation Training (SIT) | CAPS[8] | Significant time-treatment interaction [F(3, 52) = 3.08; p = 0.05, d = 1.42] in favor for NET |
| Im et al. (2018) | Pre–Post evaluation | Kenya | Somali refugees | Not specified | Trauma-Informed Psychoeducation (TIPE) delivered by lay counselors | PCL-C[9] | Significant reduction of mean for PTSD symptoms in pre–post analysis for no/low PTSD (pre: 27.42 (6.67), post: 34.48 (12.83), t(95) = −.476,p < 0.001) or high PTSD (pre: 50.09 (7.52), post: 31.93 (13.86), t(44) = 8.188, p < 0.001) |
| Knefel et al. (2022) | RCT with treatment as usual (TAU) | Austria | Afghan refugees | RHS–15[10] > 12 or RHS–15-stress scale >5 | Problem Management Plus, adapted version (aPM+) | ITQ[11] | Significant reduction of mean for PTSD symptoms in pre–post analysis for aPM+/TAU group (pre: 13.67 (4.22), post:10.96 (5.50), p = .005, dz. = 0.61) |
| Koch et al. (2020) | RCT with waitlist | Germany | Afghan refugees | Male (15–21 yrs), reporting exposure to traumatic events and difficulties in emotion regulation | Skills-Training of Affect Regulation (STARC) | PCL–5[3] | Significant Condition × Time interactions for PTSD symptoms (ΔdSTARC-Waitlist = 1.19) |
| Lancaster and Gaede et al. (2020) | Pre–post evaluation | Iraq | IDP[12] | Not specified | Resilience-based intervention (GROW) provided by paraprofessionals | SPTSS[13] | Significant reduction of mean for PTSD symptoms in pre–post analysis (t(765) = 32.22, p < .001, d = 1.16) |
| Mazzulla et al. (2021) | Pre–post evaluation | USA | Somali-Bantu and Nepali-Bhutanese refugees | Not specified | Language-free NESTT app based on cognitive behavioral and acceptance therapy techniques | RHS–15[10] | Significant reduction of mean for PTSD symptoms in pre–post analysis (t(17) = 12.23, p < .001, d = 2.83) |
| Meffert et al. (2014) | Pilot RCT with waitlist | Egypt | Sudanese refugees | HTQ[4] > 2.3 | Interpersonal therapy | HTQ[4] | Significant reduction of mean for PTSD symptoms in pre–post analysis (pre: 2.92 (0.44), post: 1.76 (0.49), difference: −1.16 (0.46); effect size for group assignment: 2.52) |
| Neuner et al. (2004) | RCT with NET[14] vs. supportive counseling | Uganda | Sudanese refugees | Not specified | Supportive counseling (SC) | PDS[6] | significant Time x Treatment interaction for PTSD symptoms in favor of NET (Wilks's λ.78), F (6,54) = 4.30, p = .01, $\eta^2$ = .31 |
| Orang et al. (2022) | RCT with waitlist | Germany | Refugees from Africa, Middle East, Arabic countries | Not specified | Value based counseling (VBC) | PCL–5[3] | Significant reduction of mean for PTSD symptoms in intervention group relative to waitlist (adjusted difference 17.15, 95% CI [10.49, 23.81], effect size 0.76, p < .001) |
| Röhr et al. (2021) | RCT with waitlist | Germany | Syrian refugees | PDS[6]-Score ≥ 11 | Sanadak App, based on cognitive behavioral therapy | PDS[6] | Nonsignificant reduction of PTSD symptoms in both groups after 4 weeks (Diff −0.90, 95% CI −0.24 to 0.47; p = .52) |

(*Continued*)

**Table 2.** (Continued)

| Author & year of publication | Study design | Country | Participants | Inclusion criteria | Stabilizing intervention | PTSD outcome measure | Outcome for PTSD symptomatology |
|---|---|---|---|---|---|---|---|
| Stanford et al. (2014) | Pre–post evaluation | Libya | IDP[12] | Not specified | HOPE-curriculum | PCL–5[3] | Significant reduction of PTSD symptoms in pre–post analysis (t(26) = 3.45, p < 0.01) |
| Stenmark et al. (2013) | RCT multicenter study with NET[14] vs. treatment as usual (TAU) | Norway | Refugees from different countries | PTSD diagnosis according to the DSM IV criteria | According to reports: help with such as sleep problems, depressive symptoms, problems related to asylum status, and other practical matters | CAPS[8] | Significant reduction of PTSD symptoms in both groups (significant main effect of time (F(2, 121.3) 30.11, p < .0001), significant main effect of treatment (F(1,71.1) 4.44, p < .05), significant Time x Treatment interaction (F(2, 122.5) 7.55, p < .001) |
| Ter Heide et al. (2011) | Pilot RCT with EMDR vs. Stabilization | Nether-lands | refugees from different countries | SCID[15] Modul PTSD and parts of the MINI[16] | Stabilization with the focus on the "here and now" | HTQ[4] | No significant of PTSD symptom reduction in either condition, but nonsignificant detorioration of PTSD symptoms for stabilization in pre–post analysis (pre:2.74 (0.27), post: 3.04 (0.25)) |
| Ter Heide et al. (2016) | RCT with EMDR vs. Stabilization as usual | Nether-lands | Refugees from different countries | PTSD diagnosis according to the DSM-IV | Stabilizing interventions according to patient's needs | CAPS[8], HTQ[4] | No significant differences between the two groups were found in either linear or quadratic slopes and effect sizes between the groups were small |
| Tol et al. (2020) | Cluster RCT with enhanced usual care | Uganda | Sudanese female refugees | Kessler 6[17] ≥ 5 | Self-Help Plus, based on ACT | PCL–6[18] | Significant reduction of PTSD symptoms posttreatment in intervention group compared to control group (mixed model analysis: −3.53 (−4.67 to −2.38), p < 0.0001 effect size = −0.68) |
| Yeomans et al. (2010) | RCT with waitlist control | Burundi | IDP[6] in Burundi | Not specified | "Healing and Reconciling Our Communities" with and without psychoeducation workshop | HTQ[4] | Significant reduction of PTSD symptoms in both intervention groups (F(2, 117) = 6.87, p < .01, partial η2 = .11), with participants in the condition without psychoeducation showing a trend for having less severe PTSD symptoms |
| Zehetmair et al. (2018) | Pre–post evaluation | Germany | English speaking refugees | ≥3 symptoms in PC-PTSD–5[19] questionnaire | Mindfulness based and imaginative stabilization techniques | PC-PTSD–5[19] | No significant reduction of PTSD symptoms in pre–post analysis (z = −1.93, p = 0.06, r = −0.47) |

Note: ALTAWIL, M., NEL, P., ASKER, A., SAMARA, M. & HARROLD, D. 2008. The effects of chronic war trauma among Palestinian children. *Children: The invisible victims of war-An interdisciplinary study: Peterborough: DSM Technical Publications Ltd.* [1]K10: Kesseler Distress Scale, 10 items,[2]WHODAS=World Health Organization Disability Assessment Schedule,[3]PCL-5: Posttraumatic Stress Disorder Checklist, [4]HTQ: Harvard Trauma Questionnaire, [5]PTSD-SRII Scale: see Altawil et al. (2008), [6]PDS: Posttraumatic Diagnostic Scale, [7]IES-R: Impact of Event Scale-Revised, [8]CAPS: Clinician Administered Scale, [9]PCL-C: PTSD Checklist – civilian version, [10]RHS-15: Refugee Health Screener-15, [11]ITQ: International Trauma Questionnaire, [12]IDP: Internally displaced person, [13]SPTSS: Screen for Posttraumatic Stress Symptoms, [14]NET: Narrative Exposure Therapy, [15]SCID: structured clinical interview, for DSM-IV Axis I Disorders (SCID-I), Modul PTSD, [16]MINI: Mini International Neuropsychiatric Interview, [17]Kessler 6: Kessler Distress Scale, 6 items, [18]PCL-6: PTSD Checklist – 6 items version, [19]PC-PTSD-5: Primary-Care PTSD-5.

Health Organization (WHO) that addresses common mental health issues. PM+ offers strategies to manage stress and addresses problems through different techniques such as relaxation, problem-solving, behavioral activation and enhancing social support (Dawson et al., 2015). This intervention modality has been conducted by both professional (Kantor et al., 2017) and peer providers (De Graaff et al., 2020, 2023), in individual and group formats (Bryant et al. 2022a,b) and across diverse living conditions (Akhtar et al., 2021; Knefel et al., 2022).

Three studies explored the efficacy of mindfulness-based interventions. These included, for instance, mindfulness-based stress reduction for refugees (MBSR-R) (Aizik-Reebs et al., 2021) and mindfulness-related and imaginative stabilization techniques that incorporated exercises such as guided imagery. The effectiveness of Acceptance and Commitment Therapy (ACT) (Tol et al., 2020), which integrates elements of mindfulness strategies with CBT, was also evaluated (Self-Help Plus) (Hayes and Pierson, 2005).

Furthermore, three studies examined stress management (SM) as an intervention approach (Carlsson et al., 2018; Barhoma et al., 2021). SM or stress inoculation training (SIT) (Hensel-Dittmann et al., 2011), both encompass relaxation techniques, attention diversion and behavioral activation. The underlying assumption is that

inadequate coping strategies may precipitate pathological stress (Lazarus and Folkman, 1984).

A similar approach was followed by three CBT-based interventions, which specifically addressed the symptom clusters of PTSD rather than general distress (Moving on after trauma, MOAT; teaching recovery techniques (TRT); HOPE-curriculum) (Stanford et al., 2014; Griggs et al., 2022; Hasha et al., 2022).

Two studies investigated interpersonal therapy as an intervention (Meffert et al., 2014; Brakemeier et al., 2017). Interpersonal therapy addresses interpersonal problems as significant contributing factors to the developement and progression of mental health impairments, including grief, role transitions, role disputes and interpersonal deficiencies such as social isolation (Lipsitz and Markowitz, 2013). The authors of the respective publications examined this therapeutic approach in more detail, arguing that both traumatic experiences, especially those of an interpersonal nature and forced displacement adversely affect interpersonal relationships and can perpetuate negative cycles (Meffert et al., 2014).

Additionally, we included two smartphone-based interventions (Sanadak app; NESTT app) (Mazzulla et al., 2021; Röhr et al., 2021). Both were based on a CBT approach, with one being language-free and incorporating ACT techniques (Mazzulla et al., 2021).

One study explored a transdiagnostic approach to improve emotion regulation skills (Skills-Training of Affect Regulation (STARC)) (Koch et al., 2020). The training has been culturally adapted to the needs of Afghan refugees based on the model of Dialectic Behavioral Therapy (DBT) (Linehan, 2014) and Skills Training in Affective and Interpersonal Regulation (STAIR) (Cloitre et al., 2010).

The remaining 11 studies examined additional approaches to stabilization, including the evaluation of different kinds of counseling. These included value-based counseling (VBC) (Orang et al., 2022) and counseling that emphasizes grieving and loss, hope and hopelessness and alienation (Alsheikh Ali, 2020). Other interventions focused on increasing resilience and dealing with feelings of helplessness (Community Wellness Focussing (CWP)) (Altawil et al., 2018), or on fostering an individual's religiousness, thankfulness, kindness, hope and courage (GROW) (Lancaster and Gaede, 2020).

Two studies investigated a manual that, in addition to psychoeducation and regulation strategies, also addresses the topics of stigma, migration stress, collective trauma (Trauma-Informed Psychoeducation, TIPE) (Im et al., 2018) and, respectively, the healing of interpersonal relationships (Healing and Reconciling Our Communities) (Yeomans et al., 2010).

Finally, three studies employed stabilization (Ter Heide et al., 2011, 2016), counseling (Neuner et al., 2004) or treatment as usual as a control condition, focusing on "sleep problems, depressive symptoms, problems related to asylum status and other practical matters" (Stenmark et al., 2008). One of the studies focused on enhancing physical safety and well-being as well as the implementation of body-oriented interventions to ease PTSD-related symptoms (Ter Heide et al., 2011). The second study focused on the enhancement of emotional regulation and the development of relational skills (Ter Heide et al., 2016). The third study employed a nonstructured counseling intervention, tailored to the patients needs with the aim of controlling for non-specific treatment effects (Neuner et al., 2004).

### Study designs

Different study designs were used to assess the effects of the interventions reviewed. Of the 31 studies, 23 were RCTs (Neuner et al., 2004; Yeomans et al., 2010; Hensel-Dittmann et al., 2011; Ter Heide et al., 2011, 2016; Stenmark et al., 2013; Meffert et al., 2014; Carlsson et al., 2018; Alsheikh Ali, 2020; De Graaff et al., 2020, 2023; Koch et al., 2020; Tol et al., 2020; Aizik-Reebs et al., 2021; Akhtar et al., 2021; Barhoma et al., 2021; Röhr et al., 2021; Acarturk et al., 2022; Bryant et al., 2022a, b; Knefel et al., 2022; Hasha et al., 2022; Orang et al., 2022), of which four were pilot evaluations (Meffert et al., 2014; De Graaff et al., 2020; Akhtar et al., 2021; Acarturk et al., 2022). Eight studies compared the intervention with a wait-list control (Yeomans et al., 2010; Meffert et al., 2014; Alsheikh Ali, 2020; Koch et al., 2020; Aizik-Reebs et al., 2021; Röhr et al., 2021; Hasha et al., 2022; Orang et al., 2022), eight further studies had an active control group, mostly compared with treatment as usual (De Graaff et al. 2020, 2023; Tol et al., 2020; Akhtar et al., 2021; Knefel et al. 2022; Acarturk et al., 2022; Bryant et al. 2022a, b). Four studies used a stabilizing intervention as an active control group for NET (Neuner et al., 2004; Stenmark et al., 2013) or EMDR (Ter Heide et al., 2011, 2016). Three studies compared two different interventions, one of which can be classified as stabilizing (Hensel-Dittmann et al., 2011; Carlsson et al., 2018; Barhoma et al., 2021). The remaining eight studies were conducted as a pre–post analysis (Stanford et al., 2014; Brakemeier et al., 2017; Altawil et al., 2018; Im et al., 2018; Zehetmair et al., 2018; Lancaster and Gaede, 2020; Mazzulla et al., 2021; Griggs et al., 2022), with one study comparing three interventions and assigning participants according to need rather than randomly (Altawil et al., 2018).

### Participant mental health burden

Inclusion criteria differed between studies in terms of the mental health burden of the study populations: 12 of the studies included refugees with a formal diagnosis of PTSD, assessed with questionnaires such as the HTQ, PDS, IES-R, PC-PTSD-5 (Meffert et al., 2014; Zehetmair et al., 2018; Alsheikh Ali, 2020; Röhr et al., 2021; Hasha et al., 2022) or described as according to ICD-10 or DSM-IV/V criteria (Ter Heide et al., 2011, 2016; Stenmark et al., 2013; Carlsson et al., 2018; Barhoma et al., 2021), or only "with PTSD" without displaying the diagnostic process or tools (Hensel-Dittmann et al., 2011; Altawil et al., 2018), or reporting exposure to traumatic events (Koch et al., 2020). One study included refugees if they met DSM-IV criteria for PTSD, affective, anxiety, somatization, eating disorders or substance abuse (Brakemeier et al., 2017). In seven other studies, refugees were included if they showed a general level of psychological distress (De Graaff et al., 2020, 2023; Akhtar et al., 2021; Acarturk et al., 2022; Bryant et al. 2022a, b; Knefel et al. 2022), as measured with the Kessler Distress Scale, Kessler 6, WHO Disability Assessment Schedule 2.0 (WHODAS 2.0) or Refugee Health Screener 15 (RHS-15). Nine studies did not specify their inclusion criteria (Neuner et al., 2004; Yeomans et al., 2010; Stanford et al., 2014; Im et al., 2018; Lancaster and Gaede, 2020; Aizik-Reebs et al., 2021; Mazzulla et al., 2021; Griggs et al., 2022; Orang et al., 2022). One study (Altawil et al., 2018) compared three interventions aimed at individuals and communities who had been exposed to severely traumatizing experiences.

### Outcome

The studies reported varying outcomes for the investigated treatment approaches (see Table 3). Nine studies showed significant reductions in PTSD symptoms for participants in the stabilizing intervention group: compared with treatment as usual, Self-Help Plus, PM+ and Mindfulness-based Trauma Recovery for Refugees

(De Graaff et al., 2020, 2023; Tol et al., 2020) showed positive outcomes. Compared with a waitlist control group, counseling that emphasizes grieving and loss, hope and hopelessness and alienation, TRTs, Skills-Training of Affect Regulation, Healing and Reconciling Our Communities and VBC (Yeomans et al., 2010; Alsheikh Ali, 2020; Koch et al., 2020; Aizik-Reebs et al., 2021; Hasha et al., 2022; Orang et al., 2022), showed favorable outcomes in six interventions. Eight studies found significant reductions in PTSD symptoms in pre–post analyses of CBT-based stabilization (MOAT), trauma-informed psychoeducation (TIPE), resilience-based intervention (GROW), peer-lead recovery group, community wellness focusing, an adapted version of PM+, interpersonal therapy or a CBT- and ACT-based (NESTT) (Meffert et al., 2014; Stanford et al., 2014; Altawil et al., 2018; Im et al., 2018; Lancaster and Gaede, 2020; Mazzulla et al., 2021; Griggs et al., 2022; Knefel et al., 2022). One study using gPM+ reported a reduction in mean PTSD symptom scores without testing for statistical significance (Akhtar et al., 2021).

Nine studies found no difference in the efficacy of a stabilizing approach in reducing PTSD symptoms: compared with treatment as usual, PM+, gPM+ did not show a better outcome (C Acarturk et al., 2022; Bryant et al. 2022a, b). When a stabilizing approach was the control condition (TAU: support with accompanying problems and practical matters), there was a significant symptom reduction in both groups compared with NET, with NET showing a greater reduction (Stenmark et al., 2013). The use of a CBT-based self-help app (Sanadak) compared with waitlist did not show significant differences in PTSD symptom reduction (Röhr et al., 2021). SM compared with cognitive restructuring also showed no significant differences in outcome, with both approaches showing nonsignificant symptom reductions (Carlsson et al., 2018; Barhoma et al., 2021). When EMDR was compared with stabilization" with the focus on the here and now" or "according to the patients needs," no significant differences between the two groups were described (Ter Heide et al., 2011, 2016), with one study reporting a nonsignificant deterioration of PTSD symptomatology for the stabilization group (Ter Heide et al., 2011). The stabilizing approach was inferior to another intervention (NET) in two studies (Neuner et al., 2004; Hensel-Dittmann et al., 2011). Two further studies assessing interpersonal therapy or the use of guided imagery reported no reduction in symptoms (Brakemeier et al., 2017; Zehetmair et al., 2018). A brief summary of the results can be found in Table 3.

### Risk of bias assessment

The results of the risk of bias assessment are presented in two separate figures: Figure 2 for randomized studies and Figure 3 for

**Table 3.** Summary of the results

| Stabilizing compared against TAU | • Three studies showing significant symptom reduction of PTSD |
|---|---|
| Stabilizing intervention against waitlist | • Six studies showing significant symptom reduction of PTSD |
| Stabilizing intervention in pre–post analysis | • Nine studies showing significant symptom reduction of PTSD<br>• Two studies showing no significant symptom reduction of PTSD |
| Stabilizing intervention in comparison with another intervention | • Nine studies no showing any difference on PTSD symptom reduction<br>• Two studies showing inferiority of stabilizing intervention in terms of symptom reduction of PTSD |

nonrandomized studies. It is evident that the majority of studies exhibit a high overall risk of bias.

### Discussion

The aim of this scoping review was to give an overview of the current literature on stabilizing, nonexposure-based interventions for refugees with trauma-related disorders. With a total number of 31 studies examined, six trials reported a significant reduction of PTSD symptom burden after conduction of a stabilizing intervention when compared to waitlist, three when compared to treatment as usual, nine in a pre–post analysis. Nine studies found no difference in the effectiveness of PTSD symptom reduction when compared to another intervention.

#### *Implemented stabilization interventions and underlying definitions of stabilization*

The question of how to define stabilization played a significant role throughout the review. Over 15 different interventions were implemented, often in varying formats (e.g., group or individual sessions, professional or lay counselors, app-based approaches) and in some

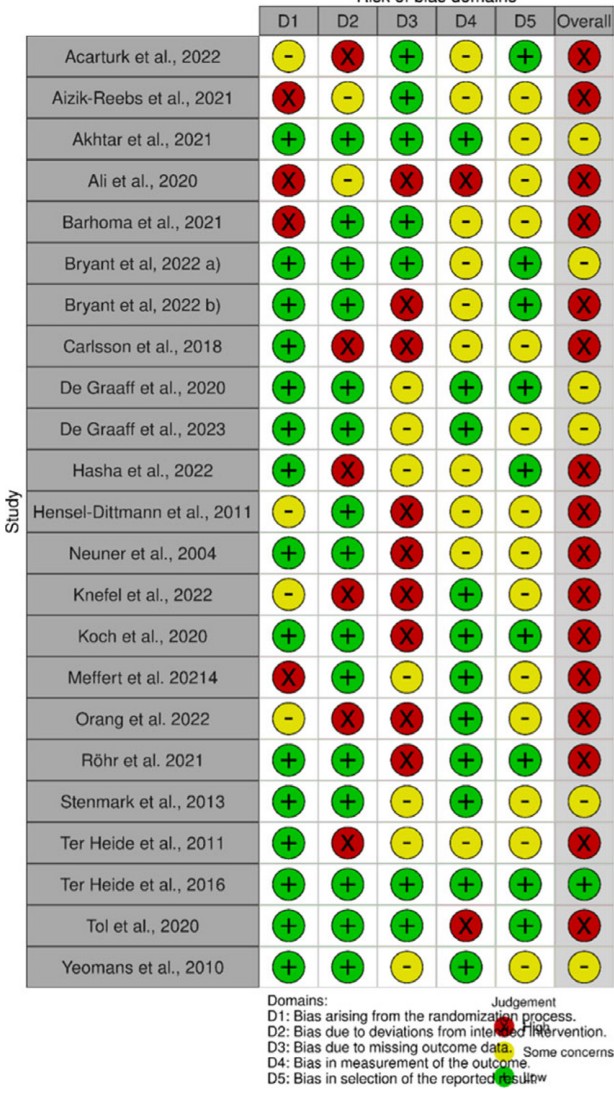

**Figure 2.** Risk assessment for randomized-controlled trials (RoB2).

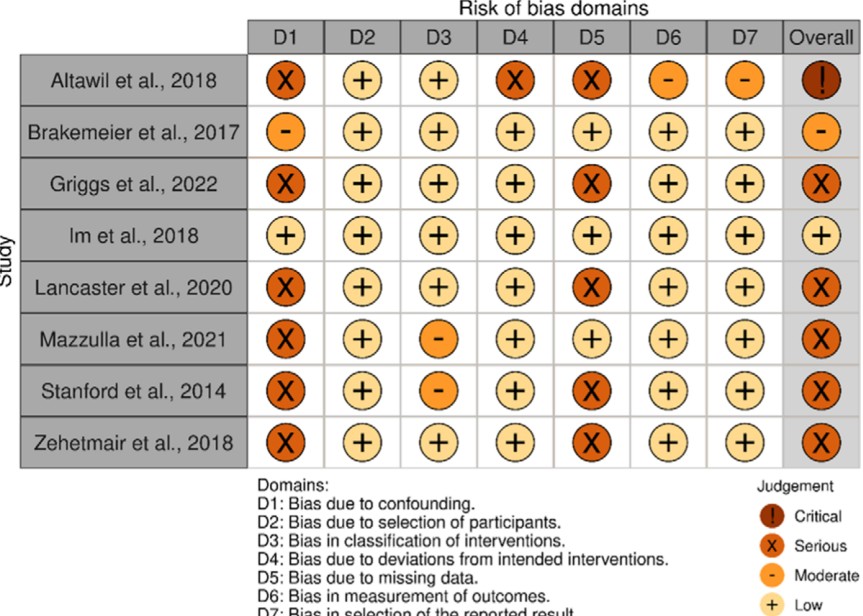

**Figure 3.** Risk assessment for non-randomized trials (ROBINS-I).

cases with limited details about the interventions. This lack of specificity made precise differentiation between the interventions challenging. It is noteworthy that, in most publications, the interventions were not explicitly described as stabilizing, though the reviewed interventions encompassed various dimensions of stabilization. Several approaches targeted emotional stabilization, employing methods such as skills training and SM to enhance individuals' capacity to manage anxiety and overwhelming emotions (Carlsson et al., 2018; Koch et al., 2020). Mindfulness-based interventions were also prominent, fostering decentering, self-compassion and reduced emotional reactivity (Aizik-Reebs et al., 2021; Aizik-Reebs et al., 2022). These can counteract typical symptoms such as hyperarousal, emotional numbness or negative mood and cognition or feelings of guilt and shame. Other interventions addressed interpersonal difficulties, which are often central to trauma-related disorders, particularly in the context of interpersonal trauma (Meffert et al., 2014; Alsheikh Ali, 2020). One study focused explicitly on rebuilding community relationships in postcivil war Burundi, emphasizing that restoring trust was pivotal in alleviating PTSD symptoms (Yeomans et al., 2010). Efforts to address everyday challenges, contributing to social stabilization, were exemplified by the Problem Management Plus intervention (Knefel et al., 2022), alongside general counseling approaches and "treatment as usual," which also provided support in navigating asylum processes (Stenmark et al., 2013).

It is possible that with a different underlying referential definition, further or different studies would have been included. For example, one study comparing social media-based drama, music and art therapy was excluded because the interventions described did not align with the definitional criteria used as a reference framework (Gever et al., 2023). On the other hand, studies were included in which psychological counseling was conducted, with one study addressing "problems related to asylum status, and other practical matters," among other aspects (Stenmark et al., 2013). This raises the question of where the boundaries between psychotherapeutic stabilization, dealing with everyday struggles and social work support lie. Existing literature highlights that prolonged asylum procedures, temporary housing and language barriers exacerbate PTSD symptoms among

refugees (Li et al., 2016; Kartal et al., 2019). Accordingly, interventions that help to deal with social difficulties could possibly have a stabilizing effect and contribute to the reduction of PTSD symptomatology. Therefore, it may be important to emphasize the aspect of social stabilization more prominently in therapeutic interventions and further investigate its impact.

### *Contextual factors influencing the effectiveness of stabilization interventions*

The overall analysis indicates that the effectiveness of the stabilizing interventions is closely associated with the participants' living conditions. Some studies found no significant effects of the stabilizing intervention on PTSD symptoms. The reasons given by the authors included the intervention's focus on addressing daily struggles, while the absence of exposure-based therapy was suggested as a factor in preventing changes in PTSD symptoms (Hasha et al., 2022). In another study, challenging environmental conditions such as poverty, separation from family, concern for their safety in the country of origin, and loneliness were identified as limiting factors for therapeutic success (Bryant, 2022; Bryant et al., 2022b), as the same intervention was successful under different living conditions (De Graaff et al. 2020, 2023). Similarly, another study hypothesized that living conditions in a reception center, combined with an uncertain residency status, contributed to the lack of improvement in PTSD symptoms (Zehetmair et al., 2018). It appears that the environmental conditions serve both as an explanatory model for why an intervention does not lead to symptom reduction but are also directly addressed as part of the therapy for stabilization. An uncontrolled study demonstrated that the number of postmigratory stressors, as well as ongoing conflicts in the country of origin, were associated with reduced symptom improvement through therapy. Additionally, an insecure residency status was linked to an increased likelihood of therapy dropout (Djelantik et al., 2020). However, a longitudinal study indicated that psychosocial interventions addressing postmigratory stressors primarily led to symptom reduction in depression and anxiety but not in PTSD (Schick et al.,

2018). Although no definitive conclusions can be drawn in this regard, the significant relevance of postmigratory stressors as an influencing factor can be acknowledged. These stressors repeatedly emerge as a critical issue in therapeutic settings (Bruhn et al., 2018).

The aspect of individual living conditions and the question of whether interventions are adapted to these conditions or even specifically address coping with them has been highlighted by many authors. The need to address daily stressors and postmigration difficulties (Knefel et al. 2022; De Graaff et al. 2023), which are known to contribute to a higher symptom burden (Gleeson et al., 2020), was emphasized. For example, Lancaster and Gaede (2020) investigated a resilience-based approach that aims to foster a person's religiousness, gratitude, kindness, hope and courage in order to help them become more resilient. This program relies on non-professional providers in a group format and posits that resilience is a necessary skill for living and surviving in a refugee camp. Moreover, the intervention problem management plus focuses on problem-solving strategies, SM, behavioral activation and strengthening social support networks. This aims to address psychosocial challenges (De Graaff et al., 2020; Bryant, 2022; Bryant et al., 2022b; De et al., 2023). It is important to note, however, that peri- and postmigratory stressors vary significantly in every context. This scoping review included studies on refugees worldwide, whose living conditions are difficult to compare. For instance, individuals living in refugee camps face different challenges than those who have arrived in a destination country and are seeking asylum. Even within the latter group, significant differences exist between those with secure residency status and those without it regarding their mental health burden (Laban et al., 2008) and effectiveness of therapeutic interventions (Ter Heide and Smid, 2015). A systematic analysis of the WHO intervention PM+ reveals that the same intervention produces highly heterogeneous outcomes across different settings (Schäfer et al., 2023). A comparable large-scale comparison is not available for other stabilization-focused studies with refugees or asylum seekers. To minimize the variability introduced by external factors, different interventions could be compared under similar circumstances rather than comparing the same intervention under different circumstances.

### Limited resources and resulting adaptations of stabilization interventions

The challenge of limited resources is a recurring issue in refugee treatment settings and frequently necessitates context-specific adaptations of interventions. Some of the respective examined interventions are easily learnable and implementable by laypersons or peers (Meffert et al., 2014; Stanford et al., 2014; Lancaster and Gaede, 2020; De Graaff et al., 2023). This highlights the challenge of limited resources, as interventions delivered by laypersons can reach a larger population compared to those requiring trained mental health professionals. It should be noted that there is also a study on NET conducted by laypersons (Neuner et al., 2008), though this is not the case for other forms of exposure therapy, such as EMDR. Additionally, other forms of resource constraints can impact the execution of studies. For example, a study conducted in a camp for internally displaced persons reported shortages of paper and printing facilities for therapy materials (Stanford et al., 2014). Group interventions also appear to be a better format for effective resource utilization. This is also reflected in the studies presented, as 15 of the 31 studies were conducted in a group format. Resource considerations also underpinned the two studies examining interventions via apps, as these are flexible and, once established,

resource-efficient. Mazulla et al. addressed another aspect by creating a language-free app to reach a larger number of people (Mazzulla et al., 2021). The current state of knowledge about smartphone-based mental health interventions for refugees was evaluated in a systematic review. The authors summarized that, up to now, none of the apps examined sufficiently met the needs of the target group (El-Haj-Mohamad et al., 2023). These factors should not only be accounted for in research studies but also considered when designing interventions for implementation outside of a research context. Not only due to limited resources but also to address shared mechanisms underlying common mental health issues, some authors have implemented transdiagnostic interventions that target PTSD alongside other conditions (Koch et al., 2020; Knefel et al., 2022). This approach was motivated by the frequent comorbidities associated with PTSD and the recognition that psychological distress among refugees extends beyond PTSD (Fazel et al., 2005; Hinchey et al., 2023a), despite the latter often being the primary focus of research (Akhtar et al., 2021; Acarturk et al., 2022; Bryant et al. 2022a, b; Knefel et al., 2022). A broader focus also allows for reaching more individuals experiencing psychological distress, which is particularly relevant in resource-limited settings, such as refugee camps or mass accommodations. Moreover, such approaches do not rely on formal diagnoses or corresponding specific interventions, making them more resource-efficient. Additionally, avoiding formal diagnostic procedures can help mitigate potential stigma, which might otherwise hinder access to effective treatment (Lancaster and Gaede, 2020). A systematic analysis of the barriers to mental health care among refugee populations also showed that it was primarily self-stigma and the fear of social consequences that prevented those affected from seeking professional help (Byrow et al., 2020). Another transdiagnostic intervention employed in refugee populations is the common elements treatment approach, which shows a significant reduction in PTSD symptom burden (Bolton et al., 2014; Bogdanov et al., 2021). However, these studies were excluded from this review because the manual for participants in low- and middle-income countries includes gradual exposure and in-vivo exposure (Murray et al., 2014).

### Needs and limitations of cultural adaptation of stabilization interventions

The studies included also repeatedly highlighted the limited applicability of Western concepts, which may not meet the needs of the target population (Altawil et al., 2018; Im et al., 2018). For SIT, the lack of cultural adaptation of the intervention for non-Western patients was also identified as a possible explanation as to why the intervention, contrary to the hypothesis, did not lead to significant symptom improvement (Hensel-Dittmann et al., 2011). The use of peers in delivering interventions could better address this aspect during the implementation of the intervention and overcome the language and cultural barriers, thus representing a lower-threshold access to psychosocial care. In some of the interventions, adaptations to the culture of the sample had already taken place within the intervention. For example, the "Community Wellness Focussing" intervention, which was carried out in Gaza, included a session on "Proverbs and Quran exercises" (Altawil et al., 2018). In a study conducted in Burundi in a camp for internally displaced persons on the other hand, the focus was on restoring social relationships within the community (Yeomans et al., 2010). Collaborative work in the community was also a major part of the study with trauma-informed psychoeducation for Somali refugees in Kenya, with sessions on "Stigma, Collective trauma, Collective healing" (Im et al., 2018). In another study, efforts were made to adapt

the intervention to the sample by consulting an advisor from the same country. In this case, gender issues and the timing of the intervention were adapted. However, no further details were disclosed (Hasha et al., 2022).

### *Implications and summary*

It becomes apparent that the authors of the presented studies provided varied responses to the challenge of addressing the complex treatment conditions of traumatized refugees, which is reflected in the heterogeneity of the study designs, interventions and results. The living conditions in which these studies were examined differ considerably. The systemic difficulties in providing psychosocial support for refugees also have an impact on the conduct of studies. The studies presented here, which were conducted under the living conditions in humanitarian settings, are subject to a variety of factors that influence the results, which must be considered in detail (Panter-Brick et al., 2020; Hinchey et al., 2023b). In 18 out of the 31 studies presented, a significant reduction in PTSD symptoms was observed as a result of the stabilizing intervention. However, a meta-analysis would be necessary to validly assess the effectiveness of stabilizing interventions for refugees, as the methods used in this review cannot provide definitive conclusions on this matter. The focus of the evaluation should consider the living conditions of refugees, as the outcomes of the same intervention may not readily translate from refugee camps to life circumstances in the country of resettlement (Acarturk et al., 2022; Bryant et al., 2022a), even though many factors, such as insecure residency status (Laban et al., 2008), family separation (Fogden et al., 2020), temporary housing (Ziersch et al., 2017; Leiler et al., 2019), unemployment (Lai et al., 2022) and language acquisition (Kartal et al., 2019), are already known to contribute to psychological distress. Furthermore, studies published in non-English languages or found in the gray literature could also be included to provide a more comprehensive and globally representative overview of stabilization approaches. Given the contextual diversity among refugee populations, qualitative studies could furthermore deepen the understanding of how refugees interpret and experience stabilization interventions in relation to their cultural, political and social realities, taking host-country-specific context into account.

### Limitations

Some limitations of this scoping review should be acknowledged: both during the screening process and the final selection of studies, stabilization was understood according to the definition provided by Luise Reddemann (Reddemann, 2011; Reddemann and Piedfort-Marin, 2017). Consequently, studies that did not align with this definition or did not offer sufficient information to ascertain their fit were excluded. With regard to the results of the bias assessment, it can be seen that limitations arise primarily due to missing data, which appear to be understandable under the living circumstances mentioned with high fluctuation and social and economic deprivation. In addition, it should be mentioned that there is a certain risk of bias in all studies, as the interventions cannot be blinded and the measurements were always patient-reported outcomes. Another important limitation should be acknowledged: among the included studies, nine did not clearly specify their inclusion criteria. In eight studies, participants were included based on a general psychological

burden without further specification, and one study also included participants with other diagnoses. Seven studies reported the presence of a PTSD diagnosis but did not provide details about the diagnostic procedures. Only six studies explicitly defined a PTSD diagnosis as an inclusion criterion and described the respective diagnostic instruments. However, the majority of the studies identified the reduction of trauma-related symptoms as their objective, even if PTSD was not defined as an inclusion criterion. A similar issue applies to the age inclusion criterion: some studies also targeted participants under 18 years old, and, in a few studies, the age of the participants was not entirely clear, leading to ambiguities in defining the target population. We chose not to exclude certain studies based on the following rationale: Refugees are frequently exposed to traumatic events, which is associated with a significantly higher prevalence of PTSD compared to the general population. Although current clinical guidelines recommend exposure-based interventions, such approaches cannot always be implemented for the reasons outlined in our introduction. A strict, linear logic – where only studies are included that demonstrate a direct, simple link between a traumatic event, a formally diagnosed PTSD using a standardized instrument, and an intervention specifically tailored to PTSD symptoms – would have led to an overly narrow selection. Such an approach would not sufficiently reflect the complexity of this research question and research environment, as elaborated in various aspects throughout the discussion. To account for this complexity and to offer a more comprehensive picture of the existing evidence, we deliberately applied broader inclusion criteria in our review.

### Conclusion

The findings of this scoping review indicate that a range of stabilizing interventions for refugees have been explored. The 31 studies that were included yielded heterogeneous results, with most showing significant PTSD symptom reduction compared to waitlist, treatment as usual or in pre–post analyses, though some found no differences between interventions, and a few reported the stabilizing approach as less effective or ineffective. Both the heterogeneity of the interventions and the environmental conditions under which the studies were conducted limit the generalizability of the results. Future studies or publications should place greater emphasis on incorporating the specific living conditions under which their results are obtained. Additionally, qualitative studies could provide valuable insights by engaging affected individuals to identify factors they perceive as contributing to symptom improvement.

**Open peer review.** To view the open peer review materials for this article, please visit http://doi.org/10.1017/gmh.2025.10028.

**Data availability statement.** Data sharing is not applicable – no new data is generated.

**Author contribution.** IR: conceptualization, data curation, investigation, formal analysis, writing – original draft, writing – reviewing & editing, NG: investigation, formal analysis, writing – reviewing & editing, JN: investigation, formal analysis, writing – reviewing & editing, BB: investigation, formal analysis, writing – rewriting and editing, HCF: resources, CN: supervision, resources, writing – reviewing & editing.

**Financial support.** This research received no specific grant from any funding agency, commercial or not-for-profit sectors.

**Competing interests.** The authors declare none.

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
