## [Reviewer Report]

General Comments on Originality and Scientific Quality:

Thank you for the opportunity to review this work. The decision to focus on and advance awareness of other effective methods of trauma recovery, in addition to exposure therapy, is appreciated and important.

Overall, the manuscript would be greatly improved by careful revision for sentence structure, grammar, wordiness, and general writing clarity/flow. As it stands, the writing is awkward at times (more of an issue in the first half of the paper), making it difficult for the reader to access the important points the authors are making. With revision for this and the comments below, I believe this will be a noteworthy contribution to the literature.

Abstract:

1. Scoping is spelled incorrectly in the title.

2. End of abstract should provide greater detail about what is meant by “favorable outcomes”; specifically, which outcomes?

3. This section of the Impact Statement is unclear and should be revised for sentence structure/grammar. “However, due to many obstacles for the implementation of adequate mental health care for refugees, stabilizing approaches might present an alternative treatment approach. This scoping review sheds light on current literature about stabilizing treatment interventions for refugees. It becomes evident, that transdiagnostic, culturally adapted interventions as well as treatment approaches conducted by lays or peers, mobile applications, and interventions that also address post-migratory stressors were examined.

Introduction:

1. The argument on page 6, lines 9-13 could be strengthened by statistics regarding drop-out rates from exposure therapy. See:

Najavits, L. M. (2015). The problem of dropout from “gold standard” PTSD therapies. F1000Prime Reports, 7. https://doi.org/10.12703/P7-43

2. Around page 7, lines 7-8, it would strengthen this argument to mention that this does not start and end with lack of clinical services, but is a systemic issue that permeates research (and lack thereof) as well. Please add this, and these and/or other citations:

Panter-Brick, M. Eggerman, A. Ager, K. Hadfield, R. Dajani, Measuring the psychosocial, biological, and cognitive signatures of profound stress in humanitarian settings: impacts, challenges, and strategies in the field, Conflict Health 14 (2020) 40, https://doi.org/10.1186/s13031-020-00286-w.

Hinchey, L., Khalil, D., Javanbakht, A. (2023). Practical approaches to conducting biopsychosocial research with refugee and internally displaced communities. Comprehensive Psychoneuroendocrinology, 16, 100217. https://doi.org/10.1016/j.cpnec.2023.100217

3. “ Attempts have been made to assess the efficacy of the standard treatments for refugees and asylum seekers, including Narrative Exposure Therapy (NET)” Here, please cite:

Kaltenbach, E., Hermenau, K., Schauer, M., Dohrmann, K., Elbert, T., & Schalinski, I. (2020). Trajectories of posttraumatic stress symptoms during and after Narrative Exposure Therapy (NET) in refugees. BMC Psychiatry, 20. https://doi.org/10.1186/s12888-020-02720-y

4. “Frequently exposed to traumatic events refugees are disproportionately affected by trauma-related disorders…” Here, please cite:

Kaltenbach, E., Schauer, M., Hermenau, K., Elbert, T., & Schalinski, I. (2018). Course of Mental Health in Refugees—A One Year Panel Survey. Frontiers in Psychiatry, 9. https://www.frontiersin.org/article/10.3389/fpsyt.2018.00352

Methodology:

1. Please state in the Methods that PRISMA was used.

2. As a meta-analysis would have provided a stronger methodology, was this approach considered? Please explain the decision to use systematic review without meta-analysis in the Methods.

Results:

1. More information is needed regarding demographics of included samples. The Results section and relevant tables should indicate the age and sex breakdown of each study sample. Additionally, it would be helpful to include a total N of participants across all included studies, as well as a sex breakdown % for the total N.

2. Importantly, as assessment of potential study bias is needed, using one or more of the available checklists to assess study bias (e.g., Cochrane Risk of Bias Tool, etc.).

Discussion:

1. Excellent point on page 16, lines 14-16. Please also cite:

Fazel, M., Wheeler, J., & Danesh, J. (2005). Prevalence of serious mental disorder in 7000 refugees resettled in western countries: A systematic review. The Lancet, 365(9467), 1309–1314. https://doi.org/10.1016/S0140-6736(05)61027-6

Hinchey, L., Nashef, R., Bazzi, C., Gorski, K., Javanbakht, A. (2023). The longitudinal impact of war exposure on psychopathology in Syrian and Iraqi refugee youth. International Journal of Social Psychiatry, 69(7), 1833–1836. https://doi.org/10.1177/00207640231177829

2. A discussion of the 9 studies that did not find significant impact from stabilizing interventions is needed. What contributed to this lack of findings?

3. Once a bias assessment has been conducted (including publication bias), limitations related to this assessment should be added to the limitations section.

---

## [Reviewer Report]

The authors provided important work, that has the potency to guide practitioners in overseeing which interventions may be suitable. However, some important limitations should be addressed before the paper is suitable for publication. I mainly focussed on the conceptual issues in this draft, and would like to share my concerns.

A clear description of what a stabilizing intervention is, would improve the manuscript. Also, it is unclear why Luise Reddemann’s concept of stabilization is chosen and how this relates to other commonly used definitions. Also in the conclusion and/or discussion more attention could be paid to what the identified interventions actually include.

The target population should be clarified. Additionally, the consequences of focusing on this group should be included in the discussion. As I understand all forced migrants worldwide are included, but the circumstances between different locations vary, and also different subgroups have varying needs, which determines the suitability of certain interventions. For example, a critical reflection on the findings in different subgroups related to the described interventions is needed.

- Rule 10: interpreters are also a concern for stabilization interventions, not specific to TFT

- Research also shows that interpreters do not diminish effectivity of TFT

- Rule 12: but when clinicians are afraid to provide TFT, should we go with their avoidance? Or should we explain to them that TFT is possible?

The rationale of the manuscript is that trauma therapy is often unfeasible (eg page 5 introduction, page 16) and therefore stabilization is a good alternative. The manuscript ignores a whole scope of literature that outlined the feasibility of trauma focussed therapy in difficult settings. References are too one-sided, and don’t do justice to the scientific efforts made to increase the accessibility of trauma therapy for people living in harsh environments. Hence, a deeper reflection in the introduction on how stabilization could suit the needs of the target population is needed. Also, outlining the current scientific gap would contribute to the relevance of the current manuscript. It is not enough, and even questionable, to state that trauma treatment is undoable. On page 5, the aim of the review is pointed out, but the concept of trauma is not included in the main aim. Hence, it is unclear what the relevance of trauma and PTSD are for the review. Page 3, line 36 mentions preceding stabilizing interventions, while later on stabilization is mentioned as an alternative. This is inconsistent. Page 4, line 17, mentions that trauma therapists are hesitant. Although this is relevant, it is not necessarily an argument for stabilization, it may also require training for therapists for example. On page 5 some examples are given that should outline the suitability of stabilization versus traumatherapy, but for example the difficulties in finding translators, or legal structures in health care provision, are not exclusive for trauma therapy and weakens the argumentation.

On page 10 two studies of Ter Heide are mentioned. Are the authors sure that they do not include the same stabilization protocol? The first study was a pilot for the second, so that would be logical. Please double check.

The discussion has a quite unclear structure and needs rigorous rewriting. On page 14, line 24, the authors state : It is noticeable that the authors’ conclusion, which are presented below, and the reasons behind carrying out the identified stabilizing non-exposure-based interventions overlap. However, the text below doesn’t describe the conclusion and is hardly related to the research question. It seems to be a repetition of the rationale (why stabilization is indicated), instead of a presentation and interpretation of the findings.

For example - Page 14, line 40, “However, a broader approach to treating psychological…. may be more appropriate for a large group of affected individuals.” Why? - Page 14, line 49, “….. does not require a formal diagnosis and corresponding specific intervention, which is also more resource-efficient” This claim is confusing, diagnosis is usually a way to optimize the match between indication and psychopathology. - Also, the claim that diagnosis is stigmatizing, line 53, “and therefore an obstacle for providing appropriate treatment” is inappropriate. - Page 14, line 60, describes the need for interventions applicable to other mental health issues, which I can only agree with. However, in this sentence the authors seem to confuse the difference between transdiagnostic and other mental health issues like depression and anxiety. - Page 15, line 26: “Another argument raised by many authors….” > Argument for what? - Page 17: Interculturality, good point, it could use some extensions. Not only peers should be included, also a reflection on adaptations in the content of treatment modules is suitable. - Page 17: The whole argument on peri-post migration stressors is not really linked to the stabilizing interventions. This section would benefit from information on what stressors are apparent and how stabilization could help, instead of pointing out that trauma therapy is hard when one deals with certain stressors. Also, the link between resilience and stabilization should be made here or earlier. The given example of deportation is important, but not necessarily applicable to all refugees. - The recommendations for future research are too shallow and should be related much more to the actual findings. What did the authors miss in the selected studies? Suggesting qualitative research, when focusing on all refugees worldwide, needs elaboration due to the huge differences between contexts. - The limitation section mentions that not only PTSD focussed interventions were included. This is problematic in relation to the rationale and final conclusion on page 17, line 60. Therefore this limitation requires more attention.

---

## [Reviewer Report]

General

This is a relevant and meaningful review! Indeed, there are hardly any studies that analyze individual therapeutic stabilization procedures and their effectiveness. Stabilizing or non-trauma-focused interventions are particularly relevant, but evidence on the efficacy is lacking. It is also a well-conducted review. Nevertheless, I have some critical comments (see below the main remarks).

Main remarks

1. Were all published peer-reviewed publications on stabilization covered in this review? I got the impression that some relevant publications were lacking, like the publications by Ethy Dorrepaal and some publications by Marilyn Cloitre. If they did not fit the search terms, make then clear why.

2. The choice of interventions is debatable. In some interventions the trauma experience was still rather central and was it simply not a real stabilization in itself (for instance, ACT and mindfulness-based intervention). Moreover, as mentioned above, important stabilization interventions appear not to be included, such as the stabilization program of Dorrepaal and Thomaes, 2009 or the Present Centered Therapy of Schnurr and others (Belsher et al, 2019; although one can argue whether this is really a stabilization approach). Deal with this issue better in the discussion section!

3. I would suggest paying more (thorough) attention to a discussion of the concept of stabilization.

4. The description of the results in the results section is very compact and not very transparent. It is difficult to read (nearly unreadable). Use specific summary tables and/or sections with subheadings.

Minor remarks

1. “Search terms related to therapy were therapy OR intervention OR treatment OR psychotherapy OR stabilization.” Was this specification, however, adequate and sufficient? Not all target interventions use the term stabilization!

2. Page 8, line 16. Here starts a very long (and highly important) paragraph. Split this paragraph into more paragraphs.

3. Page 14, line 10. “It is noticeable that the authors' conclusions, which are presented below, and the reasons behind carrying out the identified stabilizing, non-exposure-based interventions overlap”. What do you exactly mean?

4. Page 14, line 20 etc. Do not use formulations such as “Lancaster et al. (2020) highlighted” so often. It then seems like you are always hiding behind other scientists. However, this is a discussion of a scientific study and in it the authors provide their own (substantiated) interpretation of the findings.

5. Page 15, line 20. Start here a new paragraph (on apps).

6. Page 16, line 1. The paragraph on the limited applicability of Western concepts is rather incomplete. There is nowadays an interesting and relevant stream of publications on culture-sensitive approaches. It could be helpful to mention this aspect at least.

7. Page 16, line 22. “It emerges that stabilizing interventions might present an alternative to exposure-based interventions for traumatized refugees.” Not all authors would agree with this statement. Stabilization had been criticized by some as an unnecessary and even negative intervention. It is recommendable to mention this debate (see De Jongh et al., 2016).

8. The conclusion of the manuscript is more cautious that the impact statement (“While their outcomes vary, most of the examined stabilizing, non-exposure based interventions lead to significant reduction of trauma-related symptomatology”), especially because of the word ‘significant’.

References

• Belsher BE, Beech E, Evatt D, Smolenski DJ, Shea MT, Otto JL, Rosen CS, Schnurr PP. Present-centered therapy (PCT) for post-traumatic stress disorder (PTSD) in adults. Cochrane Database Syst Rev. 2019 Nov 18;2019(11):CD012898. doi: 10.1002/14651858.CD012898.pub2.

• Cloitre M, Koenen KC, Cohen LR, Han H. Skills training in affective and interpersonal regulation followed by exposure: a phase-based treatment for PTSD related to childhood abuse. J Consult Clin Psychol 2002; 70: 1067-74.

• Cloitre M, Stovall-McClough KC, Nooner K, Zorbas P, Cherry S, Jackson CL, et al. Treatment for PTSD related to childhood abuse: a randomized controlled trial. Am J Psychiatry 2010; 167: 915-24.

• de Jongh, A., Resick, P. A., Zoelner, L. A., van Minnen, A., Lee, C. W., Monson, C. M., Foa, E. B., Wheeler, K., ten Broeke, E., Feeny, N., Rauch, S. A. M., Chard, K. M., Mueser, K. T., Sloan, D. M., van der Gaag, M., Rothbaum, B. O., Neuner, F., de Roos, C., Hehenkamp, L. M. J., ... Bicanic, I. A. E. (2016). Critical analysis of the current treatment guidelines for complex ptsd in adults. Depression and Anxiety, 33(5), 359-369. https://doi.org/10.1002/da.22469

• Dorrepaal E, Thomaes K, Smit JH, Van Balkom AJ, Van Dyck R, Veltman DJ, e.a. A stabilizing group treatment for Complex Post Traumatic Stress Disorder related to childhood abuse based on psychoeducation and cognitive behavioural therapy: a pilot study. Child Abuse Negl 2010; 34: 284-8.

• Dorrepaal E, Thomaes K, Smit JH, Van Balkom AJ, Veltman DJ, Hoogendoorn AW, e.a. A stabilizing group treatment for Complex Post Traumatic Stress Disorder related to childhood abuse based on psychoeducation and cognitive behavioural therapy: a randomized controlled trial. Psychother Psychosom 2012; 81: 217-25.

• Dorrepaal E, Thomaes K, Smit JH, Veltman DJ, Hoogendoorn AW, Balkom van AJLM & Draijer N. Treatment compliance and effectiveness in complex PTSD patients with comorbid personality disorder undergoing stabilizing cognitive behavioral group treatment: a preliminary study. Eur J Psychotraumatol 2013; 4: 21171 - http://dx.doi.org/10.3402/ejpt.v4i0.21171).

---

## [Editor Report]

Thank you for submitting this paper to Global Mental Health. We have received comments from 3 reviewers who identified many strengths of the paper. They have also provided specific suggestions for improving the paper that I encourage the author to consider and incorporate into a revised version of the paper. All reviewers have identified literature that is missing from the paper (both the review and the background rationale) and have requested that the authors clarify certain points. We hope you will consider revising this paper according to their comments.

---

## [Reviewer Report]

Dear Authors,

I really appreciate your work. A lot of effort has put in the optimalization of your paper, and now it is easier to follow the line of though. Here some additional comments. The introduction is much better, it has a more modest approach which makes it easier to relate to other scientific work. I have 2 more comments for now.

1) You justify your study by pointing out low (long-term) effectiveness of trauma therapy under unstable circumstances. I believe your reasoning is not in line with the scientific state of the art. My recommendation is to choose your wording a bit more careful in relation to previous research. For example see meta-analytical findings, providing real strong evidence, like Turrini, G., Purgato, M., Acarturk, C., Anttila, M., Au, T., Ballette, F., ... & Barbui, C. (2019). Efficacy and acceptability of psychosocial interventions in asylum seekers and refugees: systematic review and meta-analysis. Epidemiology and psychiatric sciences, 28(4), 376-388. Your aim to evaluate the effects of stabilization therapy is important, and you might reconsider the strong emphasis on the alleged disadvantages of trauma focussed therapy.

2) In response to: “Thus, the aim of this scoping review is to provide an overview of the current research on stabilizing, non-exposure-based interventions for refugees through a systematic search of the literature.” (page 9) It would be good to specify this aim (as currently stated) in relation to trauma. Else the rationale, doing this review because trauma-focused therapies have limitations, is a bit incongruous. Another option is to extend on the importance of stabilization interventions, apart from the limitations of trauma therapy (that might also impact my comment 3/discussion).

Your discussion is much improved, and easier to follow for readers (or at least for me). The structure is clearer, and some core issues are pointed out convincingly. There are some -last- issues that I want to raise.

1) First some matters are a bit scattered (for example the impact of daily stressors) through the whole discussion. Improving this would increase the quality of the discussion. Also it could be helpful to add headings, to get a clearer picture of the different theme’s you point out. Another option is to clarify, in the first sentence, the point you want to make (or summarize in the last sentence the point made) in each paragraph.

2) The recommendations for future research are too shallow and should be related much more to the actual findings. What did the authors miss in the selected studies? Suggesting qualitative research, when focusing on all refugees worldwide, needs elaboration due to the huge differences between contexts. I also believe that future research also applies to how someone could do a review like yours in the future. Please elaborate a bit more on this.

3) [My first comment: The limitation section mentions that not only PTSD focussed interventions were included. This is problematic in relation to the rationale and final conclusion on page 17, line 60. Therefore this limitation requires more attention. Your response: Thank you for pointing this out. We have adjusted this aspect of the limitations in the following way „In order to gain a comprehensive picture of the study landscape, the inclusion criteria were broadly interpreted with regard to PTSD diagnosis and age. Among the included studies, nine did not clearly specify their inclusion criteria. In eight studies, participants

were included based on a general psychological burden without further specification, and one study also included participants with other diagnoses. Seven studies reported the presence of a PTSD diagnosis but did not provide details about the diagnostic procedures. Only six studies explicitly defined a PTSD diagnosis as an inclusion criterion and described the respective diagnostic instruments. However, the majority of the studies identified the reduction of trauma-related symptoms as their objective, even if PTSD was not defined as an inclusion criterion. A similar issue applies to the age inclusion criterion: some studies also targeted participants under 18 years old, and, in a few studies, the age of the participants was not entirely clear, leading to ambiguities in defining the target population. We believe through including these studies, a more comprehensive understanding of the current state of research on the treatment of trauma-related disorders among refugees can be provided.” p. 24, l. 14 – p. 25, l. 2] My current response: I don’t get the last sentence. Also, it still doesn’t reply to my previous feedback that your reason for doing this study (outlined in introduction) is in response to the limits of trauma focussed therapy (a PTSD oriented treatment) and the high prevalence of trauma exposure and PTSD among forced migrants. Hence, the argumentation of why you do this study, and focus given earlier on (You start your discussion with : The aim of this scoping review was to give an overview on the current literature on stabilizing, non-exposure based interventions for refugees with post-traumatic stress disorder.) are confusing in relation to this paragraph (especially last sentence). I think this can be solved by specifying your words a bit more.

Good luck with finishing the manuscript!

---

## [Editor Report]

Thank you for submitting the revised draft of your manuscript. One of the reviewers noted significant improvements, which I agree with and appreciate. However, they have also raised a few minor concerns related to the introduction and discussion sections. I share these concerns and encourage you to address them in a further revision.

---

## [Editor Report]

Thank you for your thorough and thoughtful response to the comments you have received from the reviewers. Your revisions to the framing and discussion have improved the clarity, rigor, and transparency of the manuscript. Overall, you have adequately addressed the key concerns from reviewers and improved the manuscript’s potential impact and contribution to the field.